# Theoretical and Experimental Investigation on a Novel Cavitation-Assisted Abrasive Flow Polishing Method

**DOI:** 10.3390/mi15091142

**Published:** 2024-09-11

**Authors:** Jiayu Wang, Xiaoxing Dong, Lijun Zhu, Zhenfeng Zhou

**Affiliations:** 1Provincial Key Laboratory of Multimodal Perceiving and Intelligent Systems, Jiaxing University, Jiaxing 314001, China; wangjiayu@zjxu.edu.cn (J.W.); dongxiaoxing@zjxu.edu.cn (X.D.); z5l5j5@zjxu.edu.cn (L.Z.); 2College of Information Science and Engineering, Jiaxing University, Jiaxing 314001, China; 3Key Laboratory of Special Purpose Equipment and Advanced Processing Technology, Ministry of Education and Zhejiang Province, Zhejiang University of Technology, Hangzhou 310023, China

**Keywords:** abrasive flow polishing, cavitation, Venturi tube, finite element simulation

## Abstract

A novel polishing method is proposed to increase material removal rates through the acceleration of abrasive movements using micro-jets formed by spontaneous collapses of bubbles due to the cavitation in a special-shaped Venturi tube. The Venturi structure is optimized by numerical simulations. Process-related parameters for the optimal cavitation ratio are investigated for achieving maximum adaptation to polishing flat workpieces. Furthermore, this novel approach enhances processing efficiency by approximately 60% compared to traditional abrasive flow polishing. The processing method that employs cavitation bubbles within a special-shaped Venturi tube to augment the flow of abrasive particles holds significant potential for material polishing applications.

## 1. Introduction

Silicon wafers, functional ceramics, and glasses, which are hard and brittle materials with distinctive physical and chemical properties, hold significant potential in the aerospace, biomedical, and semiconductor industries. Amidst escalating demands for both quantity and quality, there is a continuous emergence of novel surface polishing technologies designed for application across various production sectors. One type is the rigid contact processing technology represented by chemical mechanical polishing (CMP), gasbag polishing (GP), single-point diamond cutting, etc. [1,2,3]; the other type is the flexible contact processing technology represented by magnetorheological polishing, electrorheological polishing, float polishing, laser polishing, etc. [4,5,6,7]. In the rigid contact machining, the fixed or free abrasives are continuously subjected to normal stress to plow and remove surface materials from the workpiece. While the technique often boasts superior processing efficiency, it may also result in surface damage to the workpiece. Flexible contact polishing typically employs a fluid as a buffer medium for abrasives, which mitigates the risk of surface damage caused by the abrasives’ forceful pressure during processing. However, this method also presents the drawback of a reduced material removal rate. Thus, the prevailing challenge in current manufacturing lies in achieving a balance between efficient, cost-effective processing and the fabrication of hard and brittle materials.

Abrasive flow machining [8] uses water as the fluid medium, driving the abrasive particles to hit the workpiece at high-speed turbulence to achieve the purpose of material removal. Manoj et al. [9] explored the impact of process parameters on the abrasive water jet machining (AWJM) of Al7075 composites reinforced with TiB2 particles, employing the Taguchi-DEAR approach. Furthermore, they introduced a multi-criteria decision-making approach to optimize the process parameters in the abrasive water jet machining process [10]. E. Karkalos et al. [11] conducted AWJM experiments on a Ti-6Al-4V workpiece under diverse conditions, aiming to identify the optimal parameters for achieving a high degree of sustainability. The sustainability analysis was performed using Grey Relational Analysis (GRA), focusing on multiple indicators. These approaches enhance the scientific rigor of the abrasive erosion process on workpieces, facilitating the acquisition of optimal parameters, which in turn significantly boosts the efficiency and efficacy of the machining process. Although this low-cost polishing technology can achieve ultra-smooth surface manufacturing of hard and brittle workpieces, it inevitably faces the common problem of low processing efficiency as a flexible contact processing technology. In recent years, researchers have studied the processing characteristics of abrasive flow from different perspectives to improve processing efficiency. Zhang et al. [12] proposed using triangular constrained plate flow channels to increase the dynamic pressure of the flowing medium, enhancing the erosion performance of abrasive particles. Like the triangular constrained plate structure, constrained space was also applied to improve the efficiency of abrasive flow polishing [13]. In addition to relying on the material removal properties of the abrasive flow polishing itself, strengthening the abrasive flow with auxiliary equipment is also a method. Processing methods involving magnetic or electric field-assisted fluid viscosity variations have been applied to abrasive flow polishing. Magnetorheological fluid with self-deformable and viscoplastic abilities is used in magnetorheological abrasive flow finishing (MRAFF). Kumar et al. [14] achieved a high surface quality with a roughness average (Ra) of 0.5–2 nm in the magnetically repelling abrasive flow finishing (MRAFF) process. In addition, Zhang et al. [15] highlighted the high processing efficiency of MRAFF in the lapping application. After investigating the electromechanical principle of electrorheological fluid-assisted polishing, Fang et al. [4,16] concluded that this polishing method, which relied on the electrorheological effect, was controllable and efficient. Zhang et al. [17] proposed a liquid metal–abrasive flow machining technology under an applied electric field environment that can effectively improve the workpiece surface uniformity and increase the material removal rate. Cavitation, characterized by the implosive collapse of gas bubbles within a liquid, represents an advanced technique for material removal and the enhancement of surface properties, leveraging the energy released from the cavitation effect. Wijngaarden [18] reported that collapsing bubbles can cause metal material removal and achieve better surface quality. Chen et al. [19] proposed a cavitation water-suction polishing (CWSP) without abrasives condition to remove materials using the effect of the negative pressure cavitation method. Tan et al. [20] introduced the innovative approach of incorporating micro-/nano-bubbles into the abrasive flow polishing process, utilizing the impact force generated by the collapse of these bubbles to facilitate the erosion of the abrasive on the workpiece. Subsequently, Ge et al. [21,22] performed cavitation-assisted abrasive flow polishing using ultrasonic cavitation, effectively improving the material removal rate and achieving a high-quality surface finish.

The ultrasonic cavitation technique necessitates the use of a high-frequency ultrasonic generator, with less than 10% of its energy being harnessed for cavitation generation during the process. Consequently, there is an ongoing demand for an effective and cost-efficient method to generate cavitation. Building upon the concept of cavitation-assisted abrasive flow polishing, this paper introduces an innovative method designed to enhance the material removal rate. The approach leverages micro-jets, generated by the spontaneous collapse of bubbles due to cavitation within a special-shaped Venturi tube, to accelerate abrasive movement. The Venturi structure has been optimized through numerical simulation to confirm the parameters that yield the optimal cavitation ratio, ensuring maximum adaptability for polishing flat workpieces. Furthermore, a cavitation-assisted abrasive flow polishing system has been constructed and validated through processing experiments. These experiments demonstrate that the incorporation of cavitation in the special-shaped Venturi tube significantly improves processing efficiency.

## 2. Materials and Methods

### 2.1. Venturi Tube Analysis

The Venturi tube is a typical cavitation generating device, as shown in Figure 1a. When the incoming flow enters the flow channel’s inlet, it sequentially passes through the contraction, throat, and expansion sections. The cross-sectional area of the channel initially decreases and subsequently expands. According to Bernoulli’s equation:(1)$$p+\frac{1}{2}\rho v^{2}+\rho gh=C,$$
where *p* is the pressure of the fluid, *ρ* is the fluid density, *v* is the fluid velocity, *g* is the acceleration due to gravity, *h* is the height (or potential energy head) above a reference point, and *C* is the constant. Bernoulli’s equation illustrates that upon entering the contraction section, the fluid’s velocity increases, leading to a corresponding decrease in static pressure. Conversely, as the fluid moves into the expansion section, its velocity diminishes, and the static pressure begins to rise. Furthermore, when the hydrostatic pressure falls below the fluid’s saturated vapor pressure, cavitation is triggered within the pipeline, a phenomenon that significantly influences the flow dynamics, while the area of cavitation generated is relatively small and most of the bubbles generated inside occur in the upper part of the contraction region, where the bubbles instantly collapse. When applied to abrasive flow polishing, it is difficult for the cavitation bubbles to reach the machining area and impact on the abrasive particles near the workpiece surface. Thus, the area where cavitation bubbles are generated should be brought close to the workpiece surface by adjusting and optimizing the dimensional parameters of the Venturi structure to form the effective impact force of micro-jets to accelerate the abrasive particles. The special-shaped Venturi cavitation abrasive flow polishing structure is shown in Figure 1b.

### 2.2. Numerial Simulation Method for the Special-Shaped Venturi Cavitaiton Abrasive Flow Polishing Tools

The Venturi structure has a significant influence on cavitation results, and its size determines the proportion of the cavitation area. To obtain a larger cavitation area, it is necessary to simulate the cavitation effect of the polishing tool to obtain an optimal structure. The internal channel of the polishing tool is a three-dimensional rotator, which is simplified to a more intuitive two-dimensional model, as shown in Figure 2. According to the Venturi cavitation principle and the limitations of the actual design, the sizes of some constant parameters were determined, as shown in Table 1.

The sizes ($L_{B}\;and\;L_{C}$) and angle (α) that may have the most significant influence on the cavitation effect of the flow channel were selected through an orthogonal numerical simulation experiment. The cavitation ratio (θ) can be calculated as follows:(2)$$\theta =\frac{L_{m}}{L}\times 100\%,$$
where *L_m_* is the cavitation generation scale and *L* is the length of the processing region. The numerical calculation mainly simulated the fluid cavitation without considering the influence of abrasive particles on cavitation bubbles, and ANSYS FLUENT 18.2 version was used for the simulation. The mixture standard model was selected for the multiphase flow model, which is given as follows:(3)$$\frac{\partial \rho_{m}}{\partial t}+\nabla \cdot (\rho_{m}\vec{V_{m}})=0,$$
where $\rho_{m}$ is the mixed density and $\vec{V_{m}}$ is the average velocity of mass. The realizable k−ε standard model was selected for the turbulence model as follows:(4)$$\left\{\begin{array}{l} \rho_{1}\frac{\partial k}{\partial t}+\rho_{1}v_{1}\cdot \nabla k=\nabla \cdot \left[\left(\mu_{1}+\frac{\mu_{m}}{\sigma_{k}}\right)\nabla k\right]+p_{k}-\rho_{1}\varepsilon \\ u_{1}=\rho_{1}C_{\mu }\frac{k^{2}}{\varepsilon }\\ \rho_{1}\frac{\partial k}{\partial t}+\rho_{1}v_{1}\cdot \nabla \varepsilon =\nabla \cdot \left[\left(\mu_{1}+\frac{\mu_{m}}{\sigma_{\varepsilon }}\right)\nabla \varepsilon \right]+C_{\varepsilon 1}\frac{\varepsilon }{k}p_{k}-\rho_{1}C_{\varepsilon 2}\frac{\varepsilon^{2}}{k} \end{array}\right.,$$
where k is the turbulence kinetic energy; E is the turbulence kinetic energy dissipation rate; t is the time; $\mu_{1}$ is the dynamic fluid viscosity; $\mu_{m}$ is the turbulence viscosity; and $C_{\mu }$, $C_{\varepsilon 1}$, and $C_{\varepsilon 2}$ are the empirical constants, which are 0.08, 1.45, and 0.9, respectively. The Singhal standard model was selected as the cavitation model, which can be written as follows:(5)$$\left\{\begin{array}{c} \frac{\partial }{\partial t}(\rho )+\nabla \cdot [\rho_{\vec{v}}]=0\\ \rho =\tau \rho_{v}+(1-\tau )\rho_{1} \end{array}\right.,$$
where ρ is the equivalent density, $\vec{v}$ is the velocity of motion, τ is the gas phase ratio, $\rho_{v}$ is the density of the gas, and $\rho_{1}$ is the density of the liquid.

Meshing of ANSYS 2018 R1 was used to mesh the model structurally. The meshing was further refined in the possible cavitation regions to obtain a more accurate cavitation state, as shown in Figure 3. To ascertain mesh independence, the model’s mesh was refined to 15,812 elements for computational analysis. The results indicated that the discrepancies in each parameter, pre- and post-mesh refinement, were below 0.3%, thereby confirming that the mesh density satisfied the computational accuracy requirements. The orthogonal test, a pragmatic methodology for identifying optimal solutions, was subsequently applied. In light of the workpiece’s dimensions, the orthogonal test was conducted on three numerical simulation parameters, each at five levels, as detailed in Table 2.

### 2.3. Cavitation-Assisted Abrasive Flow Polishing Experiment

As shown in Figure 4, the polishing system was built to test the effect of cavitation abrasive flow polishing, in which the fluid can be monitored and regulated by the valve, pressure gauge, and flowmeter on the pipeline. The polishing tool is controlled by three stepper motors for self-rotation, horizontal movement, and vertical movement. The processing parameters of cavitation abrasive flow polishing are shown in Table 3.

### 2.4. Measuring Methodology

The surface of the workpiece was microscopically observed using a high-resolution scanning electron microscope, the Nova NanoSEM 450 manufactured by the FEI Corporation (Hillsboro, OR, USA). A super-depth-of-field microscope, the VHX 600 of KEYENCE Corporation (Osaka, Japan), whose observation multiple ranges from 500× to 5000×, was used to observe the surface morphology of different workpieces. The 3D morphology of four specific points in the polished area measured using Super View W1 manufactured by Chotest Technology Inc. (Shenzhen, China). The material removal rate (MRR) was calculated by the weighing method using the BSA124S analytical balance from Sartorius AG (Göttingen, Germany). In addition, the surface roughness of the workpiece was measured by the SJ-301 instrument manufactured by Mitutoyo (Kawasaki, Kanagawa, Japan).

## 3. Results and Discussion

### 3.1. Numerical Simulation Analysis

As shown in Table 4, the five levels were discretely distributed in 25 sets of numerical simulation experiments by the orthogonal method. As shown in the range analysis of the orthogonal test in Table 5, the higher the R values, the more significant the influence of the factors. Derived from the range analysis of the orthogonal test, the comparison result for the influencing factors is $\alpha >L_{B}>L_{C}$. According to the cavitation ratio (θ) in Table 4, it can be analyzed that the change in the angle (α) of the special-shaped Venturi tube leads to apparent changes in the cavitation generation area, and both smaller and larger angles result in very small cavitation areas. When α is 25°, $L_{B}$ is 1.1 mm and $L_{C}$ is 1.3 mm, Figure 5 shows that the cavitation, which may be affected by the angle (α) of the Venturi tube size structure design, does not occur in the processing area, but rather directly occurs within the contraction section.

According to the cavitation numerical simulation data from the orthogonal test, when the angle α is 55°, the size $L_{B}$ is 0.9 mm, and the size $L_{C}$ is 1.4 mm, apparent cavitation phenomena can be generated in the machining area, and the cavitation area can account for ~75.8% of the processing channel area in numerical simulation. However, the cavitation obtained from the numerical simulation of the special-shaped Venturi tube as an auxiliary research means needs to be verified by experiments. Therefore, the cavitation phenomenon is demonstrated by photographs of the observation platform. The sink made of the acrylic board at the bottom is convenient for photographing, as shown in Figure 6.

According to the analysis of the numerical simulation, the angle (α) of the special-shaped Venturi tube is the key factor affecting the cavitation ratio. Thus, the polishing tools with different angles for α were manufactured, and corresponding cavitation observations and shootings were conducted to verify the accuracy of the numerical simulation. As shown in Figure 7, the numerical simulation images of the cavitation in special-shaped Venturi tubes with different angles were compared with the actual shootings.

The white region is the bubbles generated by cavitation, showing a ring-shaped distribution, and the black region is the edge of the machining area in Figure 6. The experimental images correspond to the results of the numerical simulation. The structure of the special-shaped Venturi tube significantly affects the cavitation ratio. When the angle α is 55°, the size $L_{B}$ is 0.9 mm, and the size $L_{C}$ is 1.4 mm, an apparent cavitation phenomenon can be generated in the machining area, and the cavitation region can account for ~70% of the processing channel, which is ~5.8% different from the simulation. The difference may be related to the manufacturing precision of the polishing tools or the cavitation model selected for numerical simulation. A subtle difference in the practical tool or the simulation model would cause a dramatic change in the cavitation ratio.

In summary, the study confirms that the cavitation phenomenon can be directed towards the workpiece surface by altering the Venturi tube structure. Furthermore, the cavitation ratio can be optimized through structural design, utilizing the orthogonal test. These findings establish a foundation for the efficient processing of larger-sized workpieces in applications.

### 3.2. Experimental Analysis

The results for the silicon wafer produced by cavitation abrasive flow polishing for 2 h are shown in Figure 8a. By comparing the unpolished area, the polished area, and the sputtering-affected area, it can be seen that cavitation can accelerate material removal and gradually smooth the surface of the polished area. However, it can also be observed that the cavitation influence region obtained by numerical simulation is not the same as the polishing area showing a mirror effect. This is because the cavitation is mainly concentrated in the first half of the expansion section. By contrast, in the second half part, only the water flow drives the abrasive particles which hit the workpiece by sputtering. Thus, the material removal rate in the sputtering area is relatively slow, and the processing effect of the mirror surface cannot be presented quickly.

An ultra-depth-of-field microscope, the VHX-600 of KEYENCE Corporation (Osaka, Japan), was used to observe the surface topography of the polishing area, as shown in Figure 8b. The silicon wafer’s original surface had many pits and bumps. After the initial 2 h experiment, the surface topography of the workpiece was changed significantly, the original micro-bumps having been removed and being relatively smooth. The surface showed the characteristic erosion phenomenon of abrasive flow polishing, causing many microscopic pits, which reflected the plastic removal formed by the impact of abrasive particles. In addition, no large areas of ripple-like surface quality resulting from cavitation damage were observed after processing.

Combined with the numerical simulation of the special-shaped Venturi tube in Figure 7, it was found that the cavitation generation region is mainly concentrated above the flow channel and does not occur near the workpiece, indicating that the cavitation does not directly act on the workpiece surface. The removal of materials is caused by the impact of abrasive particles driven by high-speed water flow. Therefore, velocity flow field analysis was performed on the special-shaped Venturi tube with a larger cavitation ratio, as shown in Figure 7d. The result of the velocity flow field analysis is shown in Figure 9.

Figure 9 clearly shows the distribution of the water velocity in the flow channel, and it can be seen that the flow velocity in the region where the cavitation happens is lower. Meanwhile, the flow velocity in the area below, near the workpiece, increases exponentially compared to the cavitation zone above. This is because the angle (α) leads to a faster flow velocity near the workpiece. According to the fundamental principles of fluid dynamics, the kinetic energy of the fluid is converted into potential energy in the contraction section of the Venturi tube, and this compressed potential energy rapidly transforms back into high kinetic energy in the expansion section. The cavitation bubbles collapse swiftly within this dynamic energy conversion field, generating a significant amount of energy. The resulting micro-jets propel the abrasive particles at high velocities, increasing the likelihood of their impact with the workpiece surface. This action facilitates the acceleration of material removal in the polishing area below the cavitation zone, leading to a notable reduction in surface roughness.

To prove that the high material removal is caused by the cavitation, a comparison polishing experiment was performed on the special-shaped Venturi tube with almost no cavitation (shown in Figure 7a) and the Venturi structure with a high cavitation rate (shown in Figure 7c). After 2 h of the experiment, the comparative results of the silicon wafer surface roughness measurement were obtained, as shown in Figure 10. It can be seen from the roughness measurement results after processing that the roughness (Ra) decreased by 2.75 nm compared to the roughness (Ra) of 147.25 nm in the uncavitated area (the original surface). The roughness (Ra) with a cavitation area reduced by 19.25 nm compared to the roughness (Ra) of 128.00 nm (the original surface). In addition, the roughness of the sputtering-affected area reduced by 5.00 nm, which means that the polishing effect in this area is similar to traditional abrasive flow polishing. Furthermore, differences between groups with or without polishing were analyzed by a paired Student’s *t*-test. The extent of the effect was estimated through Cohen’s d, and statistical decisions were taken at a significance level of 0.05 or lower. The results are shown in Table 6. There were no significant differences in the uncavitated area after polishing (t(3) = 1.29, *p* > 0.05, Cohen’s d = 0.50). The area with cavitation exhibited significant differences after polishing (t(3) = 8.56, *p* < 0.01, Cohen’s d = 3.99). This difference was not significant for the sputtering-affected area before and after polishing (t(3) = 1.33, *p* > 0.05, Cohen’s d = 0.83). It is proven that the cavitation effect generated by the special-shaped Venturi design can accelerate the erosion of the abrasive flow polishing. The corresponding material removal mechanism can be derived by the numerical simulation of the special-shaped Venturi tube, as shown in Figure 11.

As shown in Figure 11, the velocity of the abrasive particles impacting the surface of the workpiece comes in two forms:Water flow directly propels abrasive particles, causing them to erode the workpiece.Due to the structure of the special-shaped Venturi tube, cavitation bubbles are generated and located higher up in the flow channel. The cavitation bubbles do not collapse directly near the workpiece surface, causing extensive cavitation erosion on the workpiece. The micro-jets generated by the collapse of the cavitation bubbles change the direction of the abrasive movement as the abrasive particles pass over the flow channel; thereby, the mechanism significantly amplifies the material removal rate, rendering cavitation-assisted abrasive erosion markedly more efficient than its traditional counterpart.

Figure 12 presents a diagram depicting the measurement points. Figure 13a,b display the respective curves and histograms illustrating the evolution of roughness and the average roughness in both the polished and sputtering-affected areas. Regardless of the processing area or the sputtering-affected area, it can be seen that the roughness decreases more significantly at the early stage of processing. Due to the numerous micro-peaks on the silicon wafer surface, the high probability of abrasive impact leads to the high efficiency of both the material removal and the roughness reduction, and the gradual smoothing of the micro-peaks in the later stages of processing leads to the lower material removal and roughness reduction rates. The roughness of the polished area was reduced below 20 nm Ra after 10 h of processing, while the roughness of the sputtering-affected area was ~60 nm Ra, a prolonged processing time still being required to reach the same roughness index.

Figure 14 shows the measurement of roughness and the material removal rate in the polished area with and without cavitation processing. It is evident from the roughness curves that cavitation-assisted polishing can improve surface roughness faster. The processing experiments with the same abrasive size (4000#) show that to reduce the roughness to below 20 nm Ra, the processing time of non-cavitation abrasive flow polishing is about 16 h, while the processing time of cavitation-assisted polishing is about 10 h. It was verified that the cavitation effect generated using the special-shaped Venturi design could accelerate the abrasive flow polishing and achieve the same surface quality with ~60% higher efficiency than traditional abrasive flow polishing. Figure 14b illustrates that the traditional abrasive flow polishing method without cavitation, utilizing SiC abrasives, achieves an MRR of 0.81 × 10^−6^ g/s. In contrast, the cavitation-assisted abrasive flow polishing method demonstrates a significantly higher rate of 1.12 × 10^−6^ g/s. This indicates that the cavitation abrasive flow polishing method outperforms the traditional method by 38.7% in terms of material removal efficiency.

### 3.3. Surface Topography Changing after Processing

Figure 15 shows the 3D morphology of four specific points in the polished area measured using Super View W1. It can be seen that the profile height of the four points is maintained in the range of 160 nm–178 nm, and the roughness is maintained in the field of 13 nm–14 nm Ra, indicating that a better uniform surface can be obtained by the erosion process using the cavitation-assisted polishing method. From the 3D morphology after processing, it can be seen that there are still impact micro-pits on the surface of the silicon wafer, which is related to the impact of the sharp SiC abrasive particles. The surface quality could potentially be enhanced by employing spherical silicon dioxide abrasives, which lack sharp angles, thereby minimizing surface damage. In subsequent studies, we plan to conduct processing experiments utilizing the cavitation-assisted polishing method with various abrasive particles. These experiments aim to explore and analyze the effects on the material removal rate and surface quality.

Figure 16 shows the SEM morphology of the wafer surface after processing. The figure shows that, whether it is the polished or sputtering-affected area, the original workpiece surface cracks or micro-peaks are removed and the processed surface becomes smoother. Compared with the morphology in the sputtering-affected area, the morphology in the polished area is similar, which can further indicate that the final processing quality is not affected by a limited amount of cavitation erosion. The translational and rotational movement of the workpiece enables the cavitation to cover the entire surface of the workpiece, which can achieve the highly efficient polishing of the whole workpiece.

## 4. Conclusions

An innovative polishing technique designed to escalate the material removal rate is proposed. This method accelerates abrasive motion by harnessing the micro-jets produced from the spontaneous implosion of cavitation bubbles within a uniquely configured Venturi tube. The essential conclusions obtained through numerical simulation and experimental analysis are as follows:Orthogonal test-based numerical simulation was employed to refine the Venturi tube’s structure, specifically tailored for the polishing of flat workpieces. When the angle α is 55°, the size $L_{B}$ is 0.9 mm, and the size $L_{C}$ is 1.4 mm in the special-shaped Venturi structure, apparent cavitation phenomena can be generated in the machining area, and the cavitation area can account for ~70% of the processing region, which is ~5.8% different from the simulation;Silicon wafer processing experiments conducted using the cavitation-assisted abrasive flow polishing system confirmed that the cavitation effect, facilitated by the special-shaped Venturi design, significantly aids the abrasive flow polishing process. This innovation has resulted in a roughly 60% enhancement in processing efficiency compared to traditional methods while maintaining equivalent surface quality standards.


## Figures and Tables

**Figure 1 micromachines-15-01142-f001:**
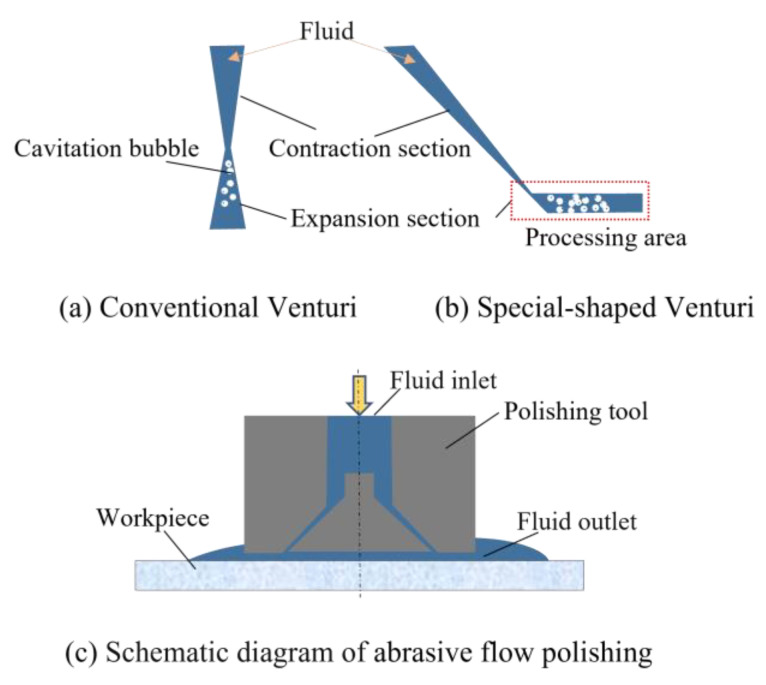
(**a**–**c**) Special-shaped Venturi cavitation abrasive flow polishing structure.

**Figure 2 micromachines-15-01142-f002:**
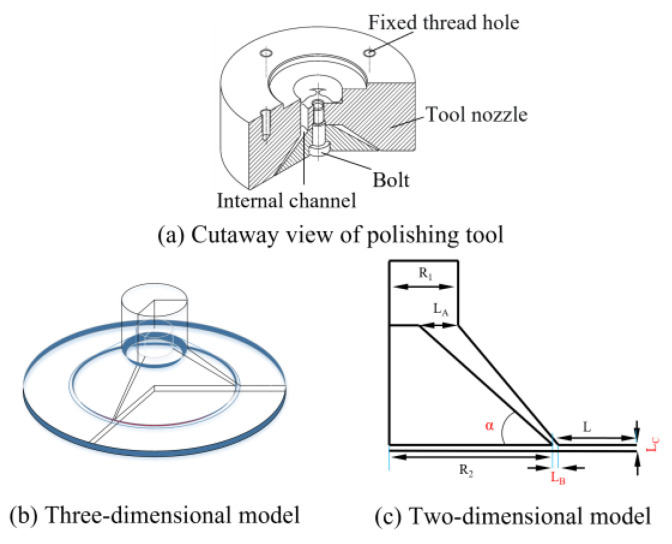
(**a**–**c**) Construction and dimensions of the polishing tool.

**Figure 3 micromachines-15-01142-f003:**
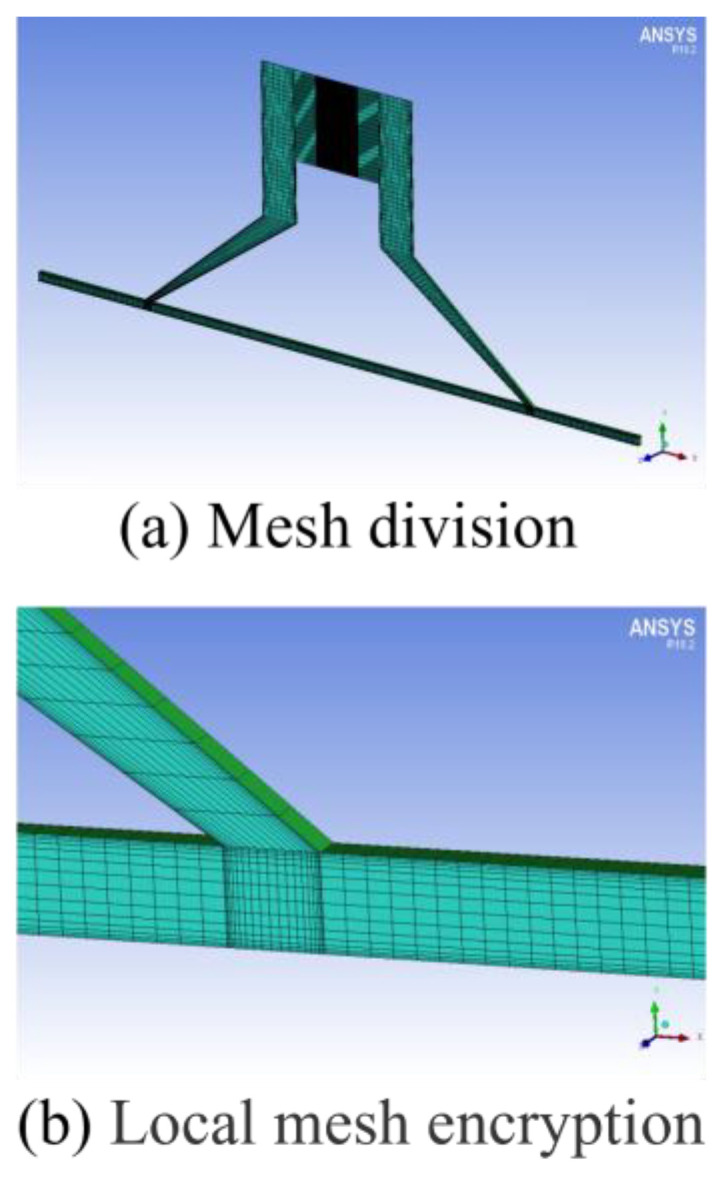
(**a**,**b**) Structured mesh generation.

**Figure 4 micromachines-15-01142-f004:**
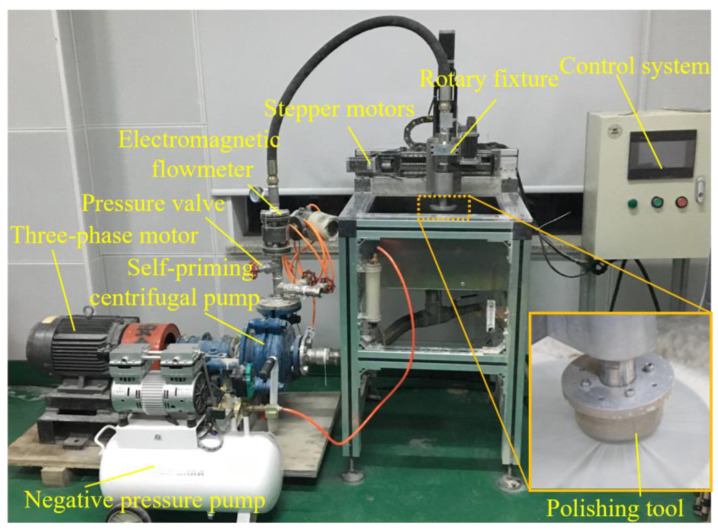
Cavitation abrasive flow polishing system.

**Figure 5 micromachines-15-01142-f005:**
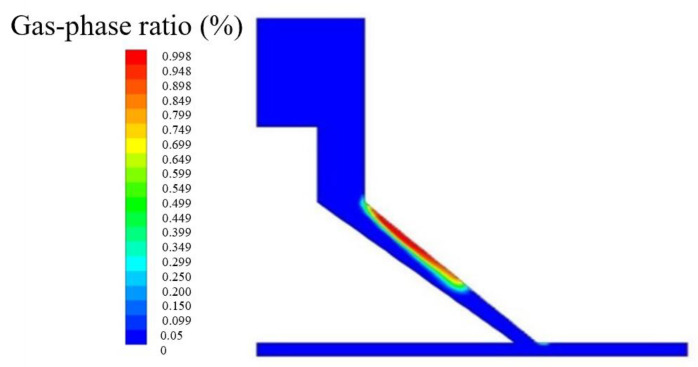
Abnormal cavitation.

**Figure 6 micromachines-15-01142-f006:**
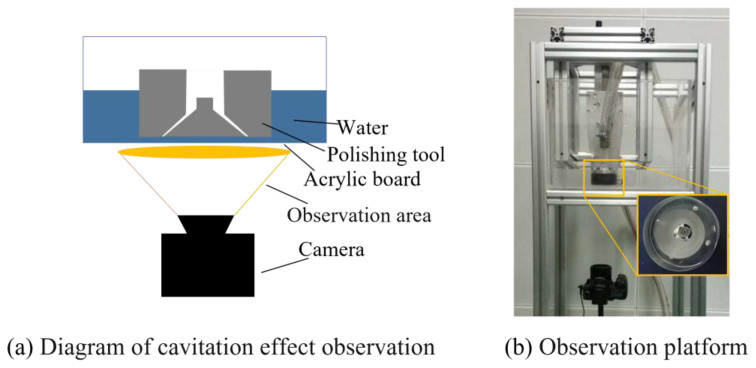
Shooting of cavitation.

**Figure 7 micromachines-15-01142-f007:**
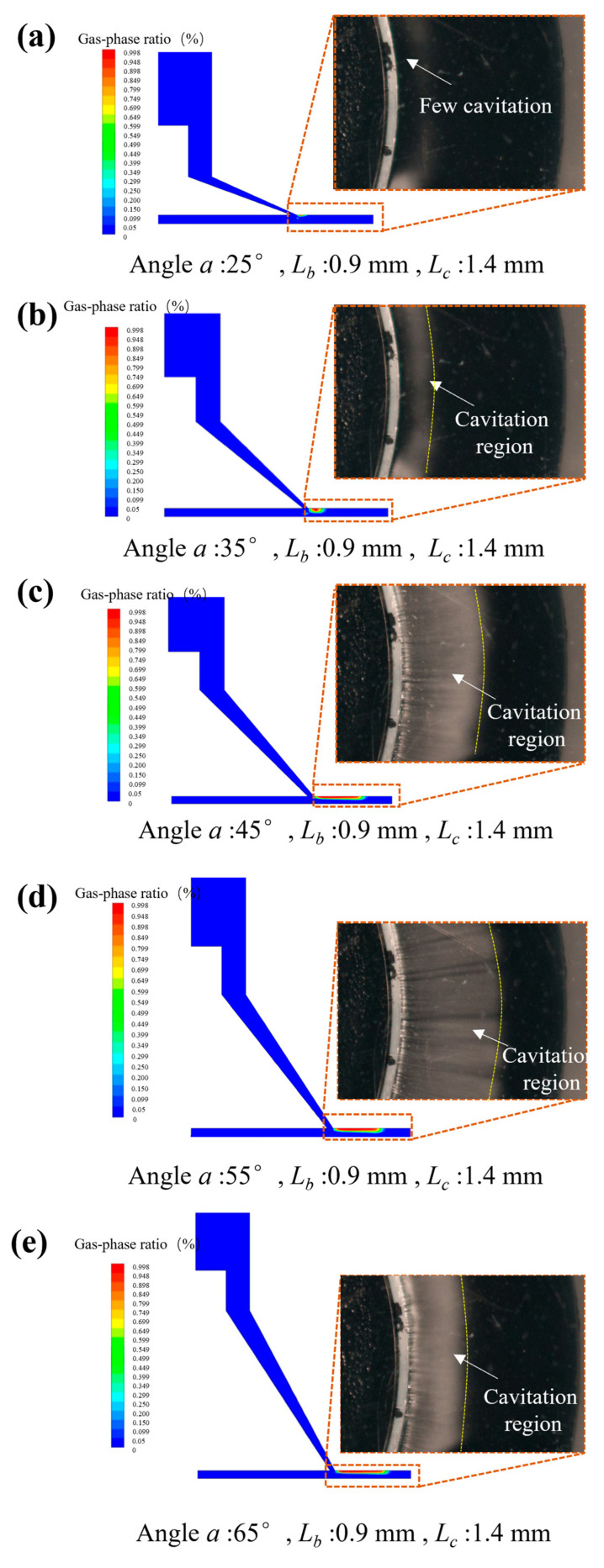
(**a**–**e**) Comparison of cavitation ratios between simulations and experiments.

**Figure 8 micromachines-15-01142-f008:**
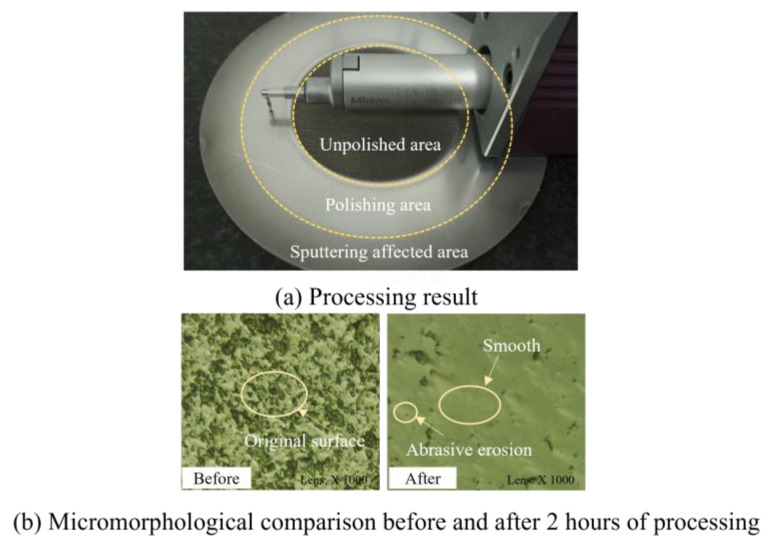
(**a**,**b**) Comparison of surface morphology before and after processing.

**Figure 9 micromachines-15-01142-f009:**
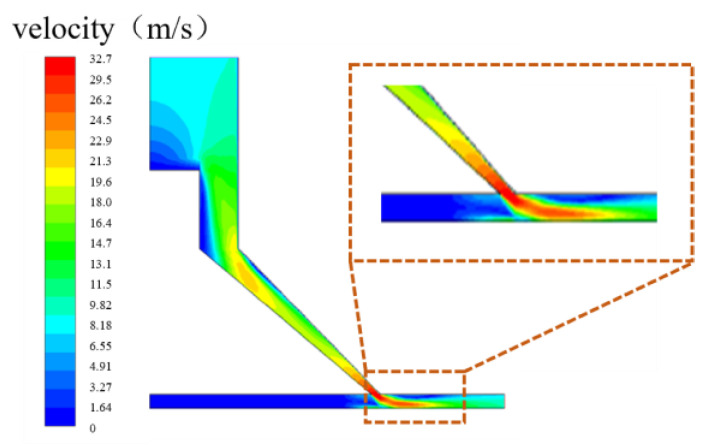
Velocity flow field analysis of the special-shaped Venturi tube.

**Figure 10 micromachines-15-01142-f010:**
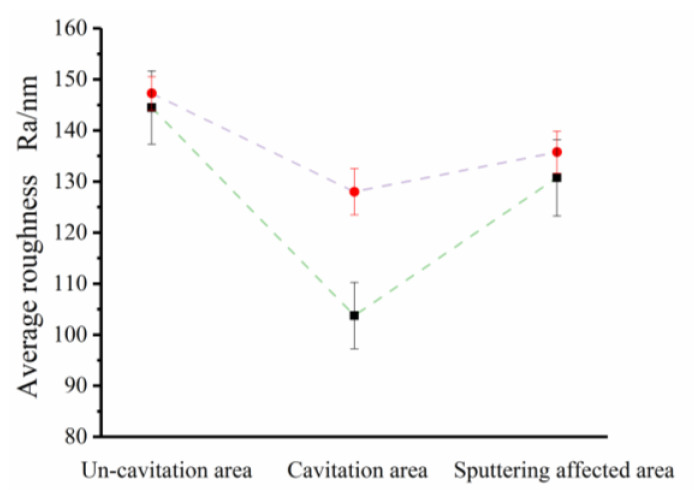
Measurement and comparison of the surface roughness of the silicon wafer.

**Figure 11 micromachines-15-01142-f011:**
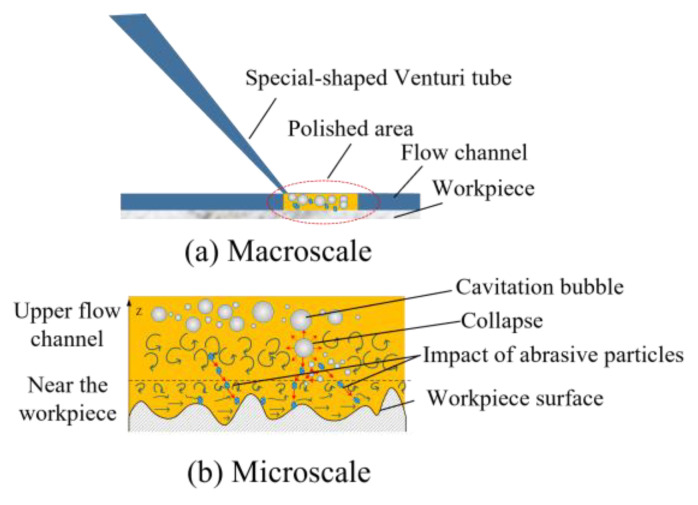
(**a**,**b**) Schematic diagram of material removal mechanism of cavitation-assisted polishing in special-shaped Venturi tube.

**Figure 12 micromachines-15-01142-f012:**
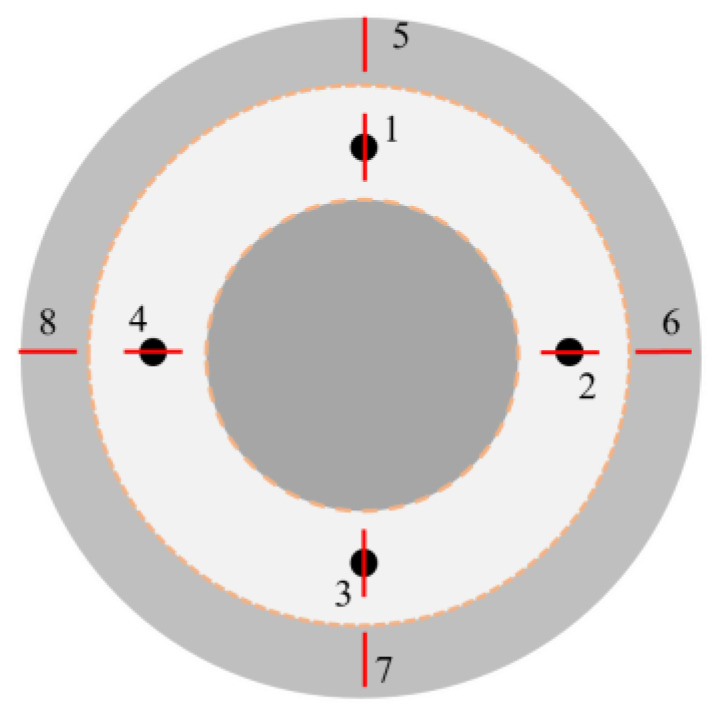
Diagram illustrating the measurement points in the silicon wafer.

**Figure 13 micromachines-15-01142-f013:**
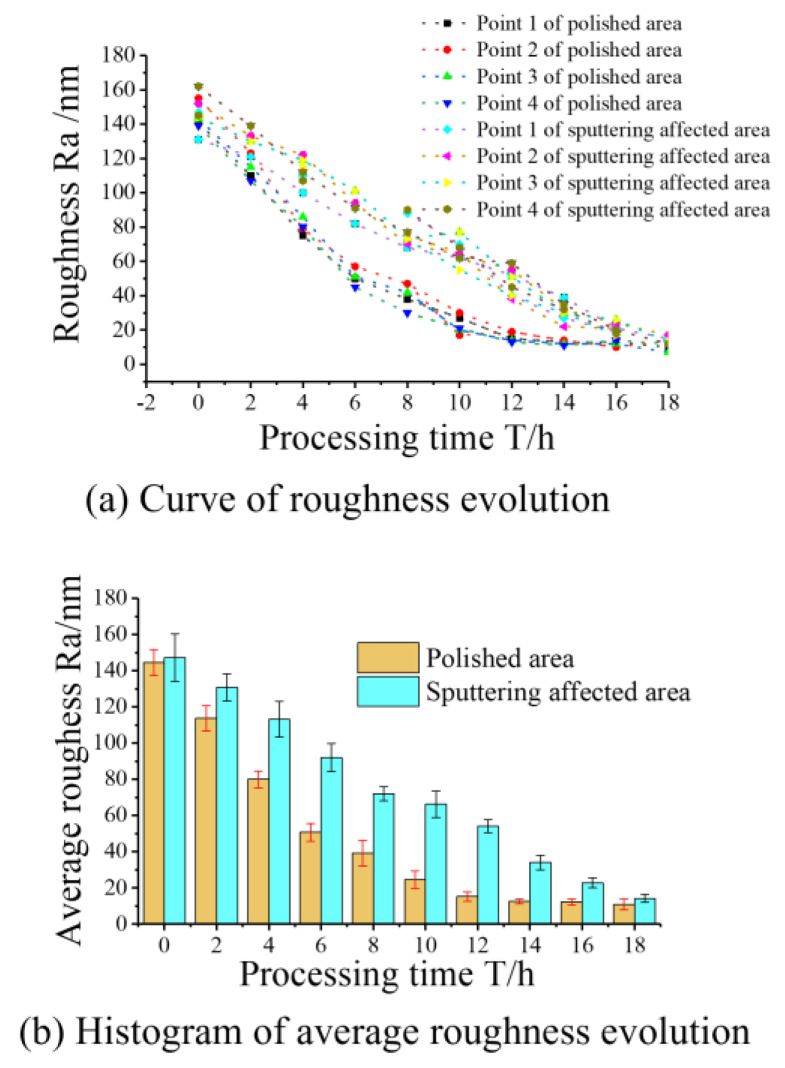
(**a**,**b**) Measurement of the roughness.

**Figure 14 micromachines-15-01142-f014:**
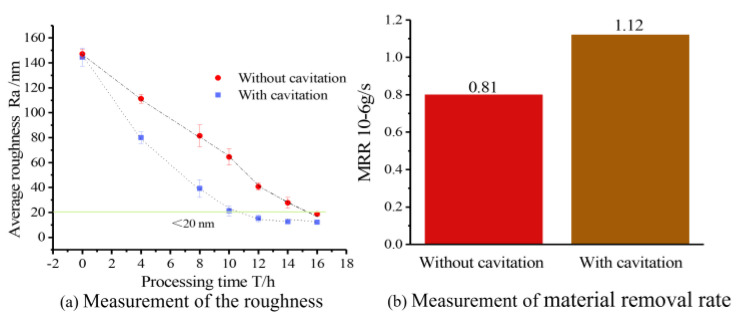
(**a**,**b**) Measurement of roughness and material removal rate in the silicon wafer.

**Figure 15 micromachines-15-01142-f015:**
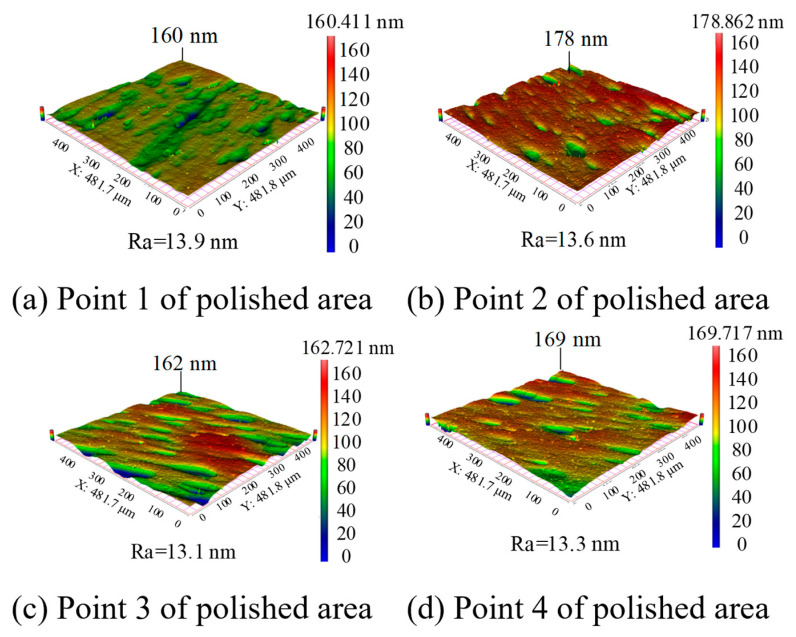
(**a**–**d**) Three-dimensional morphology of the silicon wafer surface in the polished area.

**Figure 16 micromachines-15-01142-f016:**
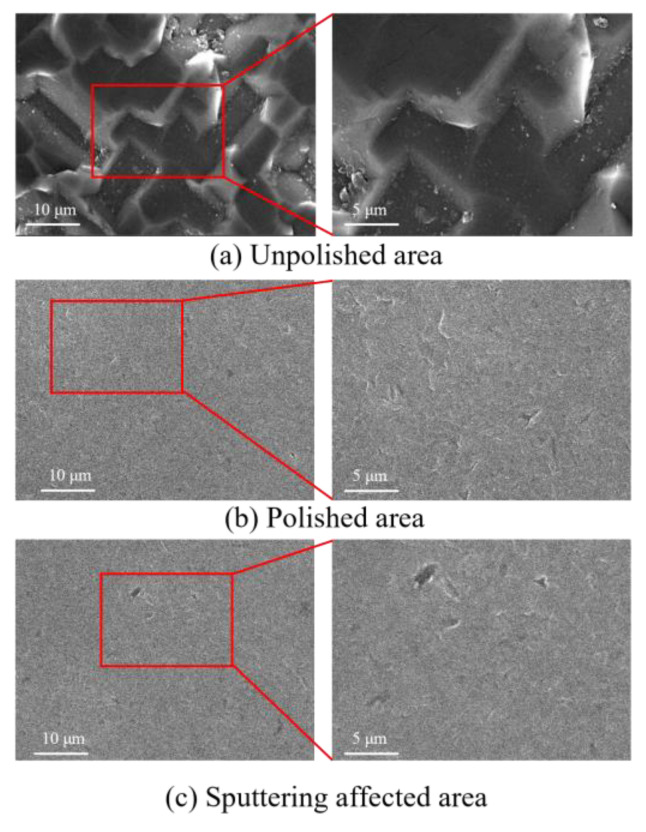
(**a**–**c**) SEM images of the silicon wafer surface after processing.

**Table 1 micromachines-15-01142-t001:** Constant parameters of numerical simulation.

Parameters	Value
Radius R1 (mm)	10
Radius R2 (mm)	40
Inlet width LA (mm)	4.4
Initial pressure P (MPa)	0.5

**Table 2 micromachines-15-01142-t002:** Orthogonal test parameters.

Parameters	α (∠)	$L_{B}$ (mm)	$L_{C}$ (mm)
Level 1	25	0.8	1
Level 2	35	0.9	1.1
Level 3	45	1.0	1.2
Level 4	55	1.1	1.3
Level 5	65	1.2	1.4

**Table 3 micromachines-15-01142-t003:** Processing parameter.

Parameters	Details
Workpiece	Silicon wafer
Abrasive	Type: silicon carbide (SiC)
	Particle size: 4000#
	Mass ratio: 10%
Base fluid	Type: deionized water; temperature: 25 °C
I nitial pressure	0.5 MPa

**Table 4 micromachines-15-01142-t004:** The results of the orthogonal test.

Parameters	α (∠)	$L_{B}$ (mm)	$L_{C}$ (mm)	Cavitation Ratio (θ)
1	25	0.8	1	2.21%
2	25	0.9	1.1	2.77%
3	25	1.0	1.2	1.47%
4	25	1.1	1.3	0%
5	25	1.2	1.4	66.16%
6	35	0.8	1.1	35.35%
7	35	0.9	1.2	26.67%
8	35	1	1.3	15.40%
9	35	1.1	1.4	65.68%
10	35	1.2	1	53.73%
11	45	0.8	1.2	28.98%
12	45	0.9	1.3	30.69%
13	45	1.0	1.4	67.57%
14	45	1.1	1	70.67%
15	45	1.2	1.1	73.65%
16	55	0.8	1.3	29.62%
17	55	0.9	1.4	75.80%
18	55	1.0	1	68.65%
19	55	1.1	1.1	62.65%
20	55	1.2	1.3	60.57%
21	65	0.8	1.4	14.60%
22	65	0.9	1	7.62%
23	65	1.0	1.1	4.72%
24	65	1.1	1.2	2.47%
25	65	1.2	1.3	2.22%

**Table 5 micromachines-15-01142-t005:** The range analysis of the orthogonal test.

	α (∠)	$L_{B}$ (mm)	$L_{C}$ (mm)
Level 1	72.61	110.78	202.88
Level 2	196.84	143.55	179.16
Level 3	271.56	157.81	120.16
Level 4	297.29	201.47	77.93
Level 5	31.63	206.33	189.85
R	869.93	819.94	769.98

**Table 6 micromachines-15-01142-t006:** Statistical hypothesis testing of roughness before and after polishing.

Condition	Group	*M*	*SD*	*T*	*df*	*p*	Cohen’s d
Uncavitated area	Before polishing	147.25	3.30	1.29	3	0.29	0.50
	After polishing	144.50	7.19				
Cavitated area	Before polishing	128.00	4.55	8.56	3	0.003	3.99
	After polishing	108.75	5.06				
Sputtering-affected area	Before polishing	135.75	4.11	1.33	3	0.28	0.83
	After polishing	130.75	7.50				

## Data Availability

The raw data supporting the conclusions of this article will be made available by the authors upon request.

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
