# Peer review of "Theoretical and Experimental Investigation on a Novel Cavitation-Assisted Abrasive Flow Polishing Method"

_micromachines, 2024, doi:10.3390/mi15091142_

Round 1

Reviewer 1 Report

Comments and Suggestions for Authors

The paper proposes a novel cavitation-assisted abrasive flow polishing method aimed at enhancing processing efficiency in polishing tasks. The authors are using experimental and computational methods. The scope of this research is quite interesting, although the authors need to make some changes before its publication. 

First, in the introduction section, the authors should clarify why this method is different from AWJ and, in general, to enrich this section. Add some information about the erosion mechanism. 

Additionally, the authors should use references to analyze the different abrasives or statistical optimization methods like th following: 

https://doi.org/10.1007/s12633-018-9763-x

https://doi.org/10.3390/su13168917

What is the novelty of your work?

The materials and method section needs to be improved. The authors should add more information about the experimental setup and the materials that they used.  Also add information about the measuring methodology and procedure  - roughness / SEM / OM etc. 

The results and discussion section does not include any discussion. The authors need to compare their findings with other relevant studies. The authors should connect the results with some physical phenomena and describe the erosion phenomena. 

The figures need to be improved according to the journal's standards. 

In Figure 9 there is no subfigure (b). 

Author Response

请参阅附件。

Reviewer 2 Report

Comments and Suggestions for Authors

Please see the report/pdf file attached.

Comments on the Quality of English Language

Extensive editing of English language required.

Reviewer 3 Report

Comments and Suggestions for Authors

This paper proposes a novel polishing method that enhances the material removal rate by utilizing micro-jets generated from the spontaneous collapse of bubbles caused by cavitation within a specially designed Venturi tube. The technique builds on the concept of cavitation-assisted abrasive flow polishing to improve surface finishing efficiency. However, there are a few improvements that could enhance clarity and readability:

1.    Section 3.2 mentions that cavitation accelerates material removal and smoothes the surface, but no quantitative measurements of material removal rate, surface roughness, or other relevant metrics are provided. The inclusion of specific data or statistics would allow for a more objective evaluation of the process.

2.    A more detailed explanation of why cavitation behaves in this way, perhaps supported by references to principles of fluid dynamics, would lead to a deeper understanding.

3.    The authors can provide a detailed explanation of the results of the velocity flow field analysis in Figure 9, which would help to understand how the fluid dynamics contribute to the material removal process. They can further explain the underlying physics that leads to an exponential increase in flow velocity and how the collapse of the cavitation bubbles affects the surrounding flow field.

4.    Row 255 to 265: The text would benefit from incorporating a statistical analysis to confirm the significance of the roughness differences and a more detailed explanation of the numerical simulation results to strengthen the connection between the experimental findings and theoretical predictions.

In general, the paper is well-written and structured. However, discussing how these results align with or differ from existing literature would enhance the credibility and relevance of the findings

Round 2

Reviewer 1 Report

Comments and Suggestions for Authors

The authors have addressed all of my comments; the paper can be accepted in its current form.

Reviewer 2 Report

Comments and Suggestions for Authors

The manuscript has been revised as per the reviewers' coments and succgestions. Therefore i would recommend the manuscript for publication.